# Whole-Process Risk Management of Soil Amendments for Remediation of Heavy Metals in Agricultural Soil—A Review

**DOI:** 10.3390/ijerph20031869

**Published:** 2023-01-19

**Authors:** Hanlin Feng, Jiemin Cheng

**Affiliations:** College of Geography and Environment, Shandong Normal University, Jinan 250358, China

**Keywords:** stabilization, amendment, heavy metal, environmental risk, whole process

## Abstract

Reducing the mobility and bioavailability of heavy metals in soils by adding exogenous materials is a technology for remediating soils contaminated with heavy metals. Unlike industrial sites, the use of such techniques in agricultural soils requires consideration of not only reducing the mobility of heavy metals but also avoiding adverse effects on soil fertility and the growth of plants. Due to the uncertainty of the stability of amendments applied to agricultural soil, the application of amendments in farmland soil is controversial. This article reviewed the field studies in which amendments were used to immobilize heavy metals, and identified the potential environmental impacts of all aspects of soil amendment usage, including production and processing, transportation, storage, application to soil, long-term stability, and plant absorption. Results of the study indicated that after identifying the environmental risks of the whole process of the application of improvers in agricultural fields, it is necessary to classify the risks according to their characteristics, and design differentiated risk control measures for the safe application of this type of technology.

## 1. Introduction

Heavy metals contamination of farmland soil is a common problem in the United States, European Union, southeast Asia, China, Japan, and other countries and regions in the world [1,2,3,4,5]. The World Food and Agriculture Organization (FAO) lists heavy metals pollution as the primary pollutant of soil [6]. Input sources of heavy metals in farmlands include atmospheric precipitation, the application of manure from livestock, sewage irrigation, sludge application, the application of chemical fertilizers and pesticides, and the storage of solid waste [7,8]. Heavy metals in soil can accumulate in the edible parts of plants through absorption by the roots, and eventually enter the human body through the intake pathway, with consequent negative impacts on health.

The use of amendments to stabilize heavy metals in agricultural soil is an in situ form of remediation technology. Broadly speaking, it is a chemical process in which an additive reacts with the contaminants in the soil to make the heavy metals less mobile [9]. As the soil to be remediated may have different uses, the stabilization targets also differ. At industrial and solid waste sites, the goal of stabilization is to reduce the risk of heavy metals leaching [10,11]. In agricultural soil, remediation materials are applied to reduce the bioavailability of heavy metals, thus reducing their mobility in the soil, and preventing their later uptake by crops. In addition, there are differences in the methods used for farmland and industrial site soil remediation. The process of farmland soil remediation requires a comprehensive consideration of soil environmental quality, soil fertility and other factors. Compared with soil stabilization methods typically used on industrial sites, the in situ remediation of heavy metals in farmlands seeks not only to reduce the effects of heavy metals on biological activity, but also to bring about improved soil fertility. Amendment is often used instead of stabilization in agricultural land remediation. The application of stabilization in the soil remediation of contaminated sites has become commonly used over the past 70 years. During this time, passivation technology has gradually become one of the most important methods used for the remediation of contaminated industrial sites [12,13]. In this regard, a certain contrast between socioeconomic development and present-day land shortages can be noted. In developed countries, brown field redevelopment can provide backup resources for urban development. Between 1982 and 2017, solidification/stabilization was used in 18.2% of U.S. superfund site remediation projects [12]. However, the application of amendments in agricultural soil is still controversial. There is as yet no unified expression for soil passivity, stabilizing agents, and stabilizing materials. Scholars often use the terms ‘amendments’ [8], ‘stabilizing agents’ [14], and ‘stabilizing materials’ [15]. As well-mixing cannot be secured in large-scale field practices, The U.S. Environmental Protection Agency (USEPA) do not use solidification/stabilization in superfund site cleaning [16]. Both the FAO and the European Union (Directive 86/278/EEC) support the use of sludge for soil improvement of agricultural land under controlled-risk conditions, but there are many restrictions on technical applications and only a limited number of amendments are permitted by law [17]. From a soil improvement point of view, it is still a technique that can be used only sparingly [18]. In some locations, the scale of heavy metal pollution of agricultural soil is large, in relative terms at least. In developing countries, the extent of heavy metals pollution of farmland soil is higher than in the European Union, the United States and other developed countries, and the area of polluted farmland is also larger. In China, the Cd content of farmland soil is 2.39 times that of the EU and 10.75 times that of Australia [19,20]. Considering the shortage of cultivated land and soil resources in developing countries, and the objective conditions associated with the large-scale restoration of cultivated land, these countries are more inclined to choose low-cost and easily accessible technologies which can be readily taken up by local residents. Such technologies are often used for the remediation of heavy metals pollution in agricultural soil in China. In 2018, the value of the Chinese remediation market was USD 2.9 billion, with solidification/stabilization the most widely used method (48.5% adoption rate). However, far fewer soil restoration projects use this technology in European countries, where the scale of individual projects ranges from 70–3000 m [21,22,23].

If the environmental impact of applied soil remediation technology is not controlled, destructive effects on the ecological environment of the soil may result, and further pollution problems may be caused by the use of amendments to immobilize heavy metals in farmland [24]. Lee et al. (2020) compared the remediation effects of land farming and high-temperature thermal desorption technologies. They found that high-temperature thermal desorption technologies caused deterioration in the fertility and ecological functions of soil [25]. Surriya et al. (2015) carried out experiments with a number of physical and chemical methods used in developing countries and found that they induced secondary damage [26]. To date, there has been no comprehensive environmental risk identification of in situ technology for the stabilization of heavy metals in farmland by means of amendments. Considering the shortage of farmland soil resources in some developing countries, and the widespread problem of heavy metals pollution [27,28,29], we believe that this low-cost in situ remediation technology has high potential market value. However, before any large-scale application of the technology is adopted, its risks should be comprehensively identified and analyzed, so that guidance for the formulation of operational specifications for applied usage can be obtained.

## 2. Soil Remediation Technology Risk Assessment Research and Application Status

### Screening Methods for Technology Applications

In the application process, the selection of soil remediation technology generally includes two procedures: preliminary screening, and detailed screening. In some countries, the preliminary screening stage involves use of the remediation technologies screening matrix [30] developed by the USEPA (Washington, DC, USA) to provide an indicator system for preliminary screening purposes (Table 1). The remediation technologies screening matrix contains 13 indicators; these include the cost of the soil remediation technology, the remediation time period, the maturity of the technology, the pollutant removal rate, the environmental impact, and the long-term effects of treatment, as well as indicators of policy compliance and social acceptance. Under actual project conditions, the engineer screens a list of alternative repair technologies in light of these indicators. For detailed screening, in the United Kingdom, Sweden and some other countries, national policy requirements also determine the available remediation technologies [31], and multicriteria decision-aid (MCDA) methods can be applied to select the remediation technology most suitable for specific contaminated soil conditions. This method is used to evaluate the advantages and disadvantages of a series of alternative soil remediation technologies by means of a constructed evaluation index system. Evaluation indicators include technological maturity, economic cost, secondary pollution, treatment effects, etc. However, evaluation index systems typically focus on the common characteristics of different technologies. For this reason, they often fail to consider the boundary conditions of different technology applications, especially the environmental risks of different technology applications and the threshold values of applicable conditions.

To date, no country or region has provided guidelines for the application of amendment in the remediation of agricultural soil. The application guidelines issued by the UK, US, and other countries all focus on the stabilization treatment of industrial or solidified-waste sites, in which the main stabilization materials used include clay minerals, lime, fly ash, phosphates, and carbonates (Table 2). The risk-control aspects of these guidelines mainly concern limitations in the application of the technology and cover such matters as the types of pollutants suitable for control and the uncertainties associated with long-term application. The Best Management Practices (BMPs) for Soils Treatment Technologies issued by the US sets out precautions to be taken before, during, and after the application of soil remediation technologies. However, these requirements are in the form of principles. No specific suggestions are given concerning the prevention or environmental risks during the application of specific technologies, or the management of long-term environmental risks. When contaminant remediation materials are used as soil amendments, the relevant guidelines specify the processing parameters, application amount, application depth, and pollutant content requirements in the materials, but they do not consider the possible environmental risks in the technical application process. Due to the lack of relevant government guidelines, the application of this technology in soil remediation has been limited. In addition, a lack of standard support in some developing countries, has resulted in the lack of a global standardization with respect to the application process of this technology, and to the environmental risks involved [39].

## 3. Risk Identification for the Whole Process of Application of Soil Stabilization Technology

There have been many studies on the remediation of soil heavy metals pollution by stabilization technology, but most of these have involved pot experiments or batch experiments. Due to the risks associated with application of this technology [46], there have been few field experiments or large-scale application studies. However, with the emergence and development of ‘green and sustainable’ remediation applications, the advantages of passivation technology in terms of its environmental friendliness, as well as its low economic cost, will likely make this a more valuable method in the future [47], especially for large-scale remediation of agricultural land contaminated with heavy metals. To date, there has been no research into risk control for the whole process involved in the passivation technology application. However, the results of field experiments enabled us to specify the environmental risks arising from each stage of the process, i.e., passivation material preparation, transportation, storage, application, and long-term monitoring. We summarized these risks in the following sections. By doing so, we hope to aid the establishment of better risk control measures for the whole process of the technology application (Figure 1) and provide support for the future application of this technology in agricultural soils.

### 3.1. Pre-Remediation

This covers the process from initial preparation to pre-application amendment preparation (Table 3). To date, a variety of passivating materials have been developed and tested, including industrial solid waste, agricultural waste, clay minerals, and synthetic materials [48]. However, some studies have considered only the efficiency of amendments in fixing to heavy metals and have ignored the potential risks to soils and crops of passivating materials containing pollutants [49,50,51]. For example, steel slag, red mud, fly ash and other industrial wastes contain heavy metals. If industrial solid wastes with a high sodium-ion content, such as red mud, are applied to soil, the pH of the soil is increased, and the biological availability of some heavy metals is reduced. However, the growth of crops in the soil is also affected [42,43]. Biochar made from crop stalks can also be used as an amendment, but often contains polycyclic aromatic hydrocarbons (PAHs) [52,53,54]. The authors of [55] found that, when organic fertilizer was used as an amendment, the mobility of arsenic in soil was enhanced, possibly through methylation, by converting arsenic into organic arsine. When farmland soil is contaminated by multiple heavy metals, especially anionic heavy metals such as chromium, or cationic heavy metals such as cadmium and lead, the application of amendments with an obvious effect on the immobilization of cationic heavy metals may activate chromium and other heavy metals, thereby increasing the risk of heavy metals pollution [56]. In addition, before the amendment is applied to the soil, it needs to be transported by fuel-powered vehicles to the soil repair site. These vehicles emit gaseous pollutants such as nitrogen oxides and volatile organic compounds. Powdered amendments are also prone to dust pollution, and often require short-term storage or processing after being transported to the site. If there are no effective anti-shower, windproofing, or anti-see-through measures in place during the storage process, the amendment itself may cause dust pollution and/or runoff pollution in rainy, snowy, or windy weather conditions. On-site crushing, mixing and other processes can also cause atmospheric or surface water pollution, if no environmental protection measures are taken.

### 3.2. Treatment

This covers the process from application of the amendments to contaminated farm soil until the time when their stabilization capacity is lost (Table 3)**.** Excessive application of amendments may affect the physical and chemical properties of soil. For example, the application of biochar in large quantities can cause acidification of the soil [57]. One study found that synthetic zeolite used as an amendment in Cd-polluted soil changed the properties of the soil, resulting in compaction [58]. Changes in the physical and chemical properties of soils can also affect the growth of crops. In addition, when a large amount of amendment is applied to soil, any amendment attached to the surface layer then enters the surrounding surface water through surface runoff, and this may cause secondary pollution problems. For example, the excessive application of phosphate amendment may lead to the eutrophication of surrounding water. Methods for mixing amendments and soils have not yet been standardized, and environmental pollution can easily be caused. The smaller the particle size of the amendment, the more easily it enters the atmospheric environment during the process of mixing with soil. For example, nanomaterials and powdered biochar materials readily enter the atmosphere when such mixing takes place on a site and can also enter the respiratory systems of construction workers [59]. In practice, if there is no adequate investigation of the hydrogeological conditions of the remediation site, the mixing depth of the soil and the amendment may exceed the depth of the groundwater level. In such a case, the amendment may cause the problem of groundwater environment pollution [60]. For example, saline-alkali soil can promote the activation of cadmium [61], and sandy soil can affect the fixation of heavy metals [62]. Such examples highlight the need for adequate analysis of the physical and chemical properties of soils on remediation sites which takes into account the findings of field experiments. Without such analysis, routine addition modification may not significantly reduce the availability of heavy metals in the soil.

### 3.3. Post-Remediation

This refers to the long-term supervisory process after the amendments have begun to fix the heavy metals in soil (Table 3). The most important reason why many countries have failed to widely apply heavy metals immobilization technology to treat farmland soil contaminated with heavy metals is uncertainty in the prediction of long-term environmental risks arising from this technology application. Many factors affect such long-term environmental risks [63,64]. Those which concern the ability of amendments to fix heavy metals in soil are relatively complex, as different types of amendments exhibit great differences in functional stability in soil. For example, Friesl et al. used an amendment to repair Cd and Pb in farmland soil. Three years later, they tested the bioavailability of heavy metals in the soil and found that the bioavailability of Cd in the soil had not increased significantly, but Pb was activated again [65]. When the amendment decomposes in the soil, the heavy metals are again converted into bioavailable states, and the impact on soil and crop safety is renewed [66]. In the practical application of this technology, it is necessary to cut off all the input channels of heavy metals simultaneously. In general, the anthropogenic input sources of heavy metals in soil include surface runoff, atmospheric deposition, and the accumulation of industrial solid wastes [67,68]. In situ remediation cannot improve the environmental quality of soil if timely measures are not taken to block the continuous input of heavy metals. For the purposes of long-term performance assessment, the influence of weather factors should always be taken into account when evaluating the chemical or physical interactions of amendments with heavy metals in soil [69,70]. The long-term stability of using biochar as an amendment was studied in one field experiment, and it was found that changes in precipitation and temperature affected the heavy-metals-absorption performance of biochar, causing a risk of secondary activation of heavy metals in soil. During the implementation of any restoration project, attention should be paid to the heavy metal absorption capacity of crops after the restoration, and the planting of crops which are targets for heavy metals enrichment should be avoided. Previous studies have shown that the uptake of heavy metals varies greatly among different crop types, and even among different genotypes of the same crop [71,72].

**Table 3 ijerph-20-01869-t003:** Environmental risks in the whole process of soil heavy metal stabilization technology.

Phase	Key Risk Factors	Method of Influence	Type of Experiment	Reference
Pre-remediation	The material itself contains contaminants	The amendment contains contaminating substances. Contaminants enter the soil with passivating agents, such as heavy metals in industrial waste and compost, polycyclic aromatic hydrocarbons (PAHs) in rice husk biochar, etc. Coal fly may contain heavy metals and organic contaminates	Field experiment	[73,74,75,76]
		The amendment has other potential environmental risks, such as the self-toxicity of nanomaterials. Biochar pyrolysis can damage crop cell structure and produce aromatic compounds, which are easily absorbed by crops	Field experiment	[77,78]
		The physical and chemical properties of the amendment may affect soil production function. For example, red mud applied to the soil significantly increases soil pH and affects crop growth		[79]
		The amendment can activate other nontarget contaminants. For example, organic fertilizer can activate heavy metals such as arsenic in soil through methylation reaction. Passivating materials increase soil pH and organic matter content and activate arsenic contaminants in soil	Pot experiment	[80,81,82]
	Amendment unsuitability for compound contaminated soils	When polution involves a cationic heavy metal and arsenic together, it is necessary to consider that the activity of arsenic will rise with increases in pH, with the opposite stabilization effect to that of the cationic heavy metal. For example, in cases of contamination with both cationic heavy metals and arsenic, the use of phosphate as an amendment may activate arsenic.	Field experiment	[83]
		The amendment has a limited stabilizaiton effect on specific heavy metals, as with the use of plaster to repair Cr(III)-contaminated soil, which is easily converted to more toxic and mobile Cr(VI) under neutral conditions.	Batch experiments	[84]
	Material manufacturing processes emit pollutants	Manufacturing processes produce pollutants. For example, passivating agents enter the atmospheric environment through dust and volatile organic compounds	/	/
		The particle size is too large, the stabilization effect is not obvious, and the risk of heavy metals to crops is not alleviated. Excessively large particles also affect the physical and chemical properties of soil	Pot experiment	[85]
		The particle size is too small, and the amendment has strong migration. For example, nano-passivation materials more readily diffuse in the atmospheric environment, and penetrate vertically into the groundwater environment with greater ease. Passivated materials with too small a particle size readily carry heavy metals to the groundwater or diffuse into the atmosphere in the form of dust	Field experiment	[86,87,88]
	Discharge of pollutants during material transportation	Emission of pollutants such as exhaust gas from transport vehicles. A lack of necessary protective facilities during transportation causes the amendment to enter the atmosphere as dust and as volatile organic compounds	/	/
	Discharge of pollutants during material storage	Due to a lack of necessary protective facilities during storage, the passivating agent enters the atmosphere as dust and as volatile organic compounds. The amendment enters the surface water environment through surface runoff, or the passivation agent enters the soil environment through leakage	/	[57]
Remediation treatment	Application rate	An excessive application rate affects the physical and chemical properties of soil, causing acidification and hardening, and affecting crop growth. For example, nano-passivation materials affect soil aggregate structure and nutrient availability, and their excessive use reduces crop yield	Field and pot experiments	[66,77,89,90]
		An excessive application amount and an excess of passivation material cause secondary pollution. For example, an excessive amount of phosphate amendment applied causes phosphorus loss	Field experiment	[91,92]
	Application process	Biochar and nanomaterials can easily drift into the air and affect human health through breathing	Field experiment	[93,94]
	Application depth	The application depth is too deep, and the groundwater environment is affected	/	[95]
		The application depth is too shallow, the amendment acts on the non-rhizosphere soil, and cannot inhibit the uptake of heavy metals in rhizosphere soil	Field experiment	[15,57]
		Tillage depth is too deep, leading to the vertical migration of soil surface pollutants	Field experiment	[74]
		The application depth is too shallow, the amendment fails to enter the soil environment, but enters the surface water environment through surface runoff, or diffuses into the atmospheric environment	Field experiment	[96,97]
	Effect of soil properties	The physical and chemical properties of the soil may affect the passivation effect; for example, salinized soil increases the activity of Cd. If the soil particle size is too large (as in sandy soil), the passivation effect is poor	Pot experiment	[61]
Post-remediation	Decay of amendment properties	Degradation and transformation of the amendment, failure of passivation, and re-activation of heavy metals affect soil and crop safety	Field experiment	[98,99,100]
		The amendment decomposes due to aging, then migrates and transforms in the soil, affecting soil and crop safety	Field experiment	[56,69,74,101,102]
		The stabilization effect of clay mineral amendment is significantly reduced after long-term use. A three-year field experiment showed that Cd passivation is not significantly reduced after sepiolite and limestone passivation, but Pb passivation is significantly reduced	Field experiment	[101,102]
	Meteorological conditions	Precipitation and temperature changes lead to the weakening of stabilization effect and to the reactivation of heavy metals	Field experiment	[69]
	Continuous input of pollutants	Pollutants are continuously imported into soil through surface runoff, atmospheric deposition, or solid waste dumping	Field experiment	[103,104]
	Effects of heavy metal uptake by crops	After the application of the amendment, the target heavy metals in edible parts of crops are reduced, but other heavy metals may increase	Pot experiment	[105]
		No significant reduction in the target heavy metal content in the edible part of the crop	Field and pot experiments	[106,107]
		Special types of crops have the ability to enrich heavy metals	Field and pot experiments	[61,97,98]
	Risk of combined technology application	The secondary environmental pollution caused by the loss of passivated materials with surface runoff can be avoided when the amendment is combined with flooding, to remediate cadmium-contaminated rice fields	Field experiment	[108,109,110]

## 4. Conclusions

The use of amendments for the remediation of agricultural soils polluted by heavy metal is a green and economical technology. Due to the lack of necessary risk prevention technical standards for the technology application, there may be some environmental risks in the use of amendments. In some countries, especially in China, there are various types of amendments which have been used in field or studied in the laboratory. Since the combination of amendments and heavy metals in the soil is mainly a chemical process, the mobility of the amendments and heavy metal in the soil-plant system could pollute the soil, groundwater, air and plants. Therefore, such technologies need to be used with caution. The premise of the safe application of the technology is to establish a risk control system for the whole process of technology application, and to take measures to control environmental risks after identifying and classifying different stages and different types of environmental risks in the whole process of the technology.

(1)Pre-remediation. Screening of amendments should be carried out before technology application to avoid the introduction of environmental pollutants and hazardous substances into the soil, especially some emerging amending materials should be paid attention to, and large-scale application in farmland is not recommended until its safety is proven through field trials. It is necessary to control the secondary pollution during the manufacturing, transportation, and storage of amendments, especially in countries where remediation projects will be carried out on a large scale. These remediation projects are particularly concerned with dust, VOC pollutants, and carbon emissions.(2)Treatment. The hydrogeological conditions of the site to be remediated should be evaluated in the technical application to avoid contamination of the surrounding water and groundwater. Since the object of the technology application is agricultural soil, the impact of the physicochemical properties of the amendments on soil physicochemical properties and soil microorganisms should also be evaluated in the technology application. It is recommended to establish the acceptance criteria of the completed project, including the pollutant content of crops, crop yield, soil fertility, and surrounding environmental quality, all of which should be included in the assessment index system.(3)Post-remediation. After the application of amendments, it is recommended to carry out long-term monitoring, including monitoring of the yield and quality of crops, and the environmental quality of surface water and groundwater around farmland, to ensure the safety of technology applications. The bioavailability of heavy metals in the soil should be kept at a low level by monitoring and recommending timely remedial measures when amendments are found to have failed. The bioavailability of heavy metals in the soil should be kept at low levels by monitoring and recommending timely remedial remediation measures when the immobilization effect of the amendment has decreased.

## Figures and Tables

**Figure 1 ijerph-20-01869-f001:**
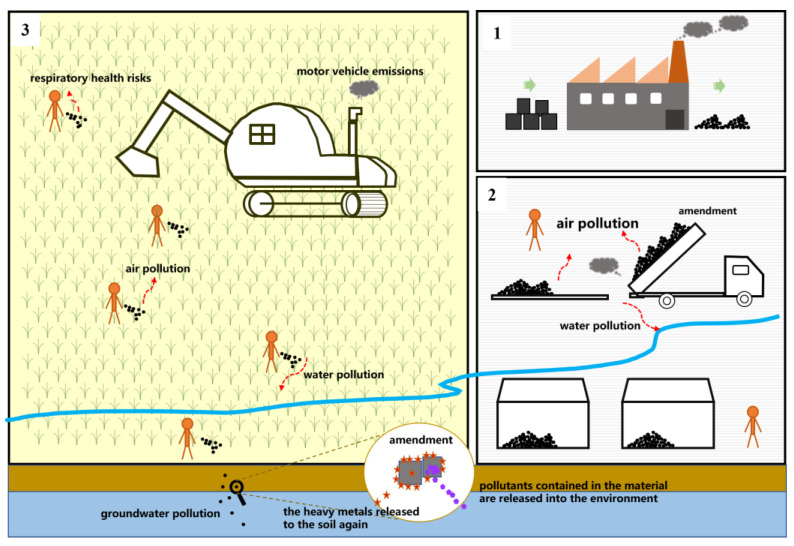
Conceptual model of environmental risk in the whole process of application of soil heavy metal pollution amendments. (**1**) amendment production; (**2**) amendment transportation and storage; (**3**) amendment application. Purple dots represent the pollutants contained in the amendment, orange asterisks represent the heavy metals that is fixed by the amendment.

**Table 1 ijerph-20-01869-t001:** Technical screening methods for some farmland restoration projects.

	Application Area	Type of Site (Major Contaminants)	Methodology for Selecting Technologies	Advantage	Disadvantage	References
1	Porto Margheta, Venice, Italy	Industrial site (heavy metal and benzene serial materials)	Screening matrix developed by USEPA (MATRIX)-MCDA	According to the characteristics of contaminated soil to be remediated, the suitable technology is determined by combining qualitative and quantitative methods	The dimension of environmental risk assessment is single, and gaseous pollutants (dust, VOCs, etc.) are more substantially considered	[32]
2	Beijing, China	Industrial site (PAHs and BTEX)	Monte Carlo analytic hierarchy process (MAHP)	A relatively comprehensive analysis is obtained	Human interference factors have a great impact	[33]
3	China	Agricultural site (organochlorine)	Analytic hierarchy process (AHP), Technique for Order of Preference by Similarity to Ideal Solution (TOPSIS)	Detailed technical and economic parameters of different technical applications are given	Environmental risk factors are less considered	[34]
4	Tianjin, China	Industrial site (not provided)	LCA-AHP	A range of pollutants, such as greenhouse gases, waste gas, waste water, and noise are considered	Risk assessment indicators are not well targeted	[35]
5	Iowa, USA	Lake sediments (PCBs and pesticides)	LCA-AHP	A variety of factors, including greenhouse gases, solid waste, energy, exhaust gas, wastewater, and health risks are considered	No obvious disadvantage	[36]
6	Wuhan, China	Industrial site (volatile organic contaminants and heavy metal)	Input–output life cycle assessment	The environmental impacts of different technologies are quantified	Only four gaseous pollutants, and few environmental factors, are considered	[37]
7	Tianjin/Liuzhou/Linyi, China	Industrial site (organic contaminant)	Comprehensive evaluation method framework	Greenhouse gases, sulfur dioxide, ozone, odor, eutrophication, and noise are considered	Concentrates on the implementation stage of restoration projects, and does not adequately consider environmental risks in the preparation and after-restoration stages	[38]

**Table 2 ijerph-20-01869-t002:** Selection methods for soil remediation technologies in some countries.

Country or Region	File	Amendment (Stabilization)	Environmental Risk Prevention Measures	References
The International Biochar Initiative	Standardized product definition and product testing guidelines for biochar that is used in soil	Biochar	Toxicant assessment of biochar materials	[40]
The UK Environment Agency	Guidance on the use of stabilization/solidification for the treatment of contaminated soil	Lime	Control of quality of stabilization materials and of pollution during the technology application	[41]
USEPA	Best Management Practices (BMPs) for Soils Treatment Technologies	Phosphates, sulfides, carbonates, etc.	Environmental protection measures before, during and after the implementation of the project are stipulated. However, the pertinency is insufficient, and no long-term follow-up assessment measures are proposed for some technical applications	[42]
The Environmental Protection Department of Hong Kong	Practice guide for investigation and remediation	Cement	The limitations of different repair techniques are listed, but no targeted risk prevention measures are proposed	[43]
USEPA	Handbook for stabilization/solidification of hazardous wastes	Bottom ash, fly ash, lime, zeolites, clay	The limitations of different repair techniques are listed, but no targeted risk prevention measures are proposed	[44]
Washington State Department of Ecology	Guidelines and Resources for Implementing Soil Quality and Depth BMP T5.13 in WDOE Stormwater Management Manual for Western Washington		A modification of the contaminant content is proposed	[45]

## Data Availability

Not applicable.

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
