# Peer review of "Whole-Process Risk Management of Soil Amendments for Remediation of Heavy Metals in Agricultural Soil—A Review"

_ijerph, 2023, doi:10.3390/ijerph20031869_

Round 1

Reviewer 1 Report

(1) In the paper, when talk about stabilization, the authors used amendment, stabilizaiton, fixation. I suggest the authors introduce the difference between amendment and stabilizaiton.

(2)In” 3. Risk Identification of the Whole Process Application of Soil Stabilization Technology” , supple the definition of each phase scope.

(3)In 3.1,there is no evidence for the poin of biochar contains PAHs.

(4)In Table 3, it is useless to edit numbers for each soil stablization process.

(5) Whether any country or region has made requirements for the whole-process risk control of soil remediation technology through laws or technical specifications? If necessary, provide it in the background section

(6) It is suggested that in the conclusion part, the authors can give suggestion for the countries to carry out the management of agricultural land soil restoration, so as to ensure that the environmental risks in the passivation restoration process are controlled.

Author Response

Response to reviewer 1.

(1) In the paper, when talk about stabilization, the authors used amendment, stabilizaiton, fixation. I suggest the authors introduce the difference between amendment and stabilizaiton.

Response:

Compared with the use of soil stabilization in the industry sites, the in-situ remediation of heavy metals in farmland soil should not only re-duce the biological activity of heavy metals, but also consider the improvement of soil fer-tility. In research and application, the term soil amendment is often used(Lines 42-48).

(2)In” 3. Risk Identification of the Whole Process Application of Soil Stabilization Technology” , supple the definition of each phase scope.

Response:

The paper supplied the description of different stages of soil stabilization technology(Lines 142-143, 168-169, 192-193).

(3)In 3.1,there is no evidence for the poin of biochar contains PAHs.

Response:

The paper supplied with three articles on the evidence of PAHs in biochar(Lines 52).

(4)In Table 3, it is useless to edit numbers for each soil stablization process.

Response:

We removed the ordinals in Table 3(Lines 220).

(5) Whether any country or region has made requirements for the whole-process risk control of soil remediation technology through laws or technical specifications? If necessary, provide it in the background section.

Response:

At present, there is no evidence that any countries and regions have established relevant laws, regulations or standards for risk prevention and control in the whole process of soil remediation technology application.

(6) It is suggested that in the conclusion part, the authors can give suggestion for the countries to carry out the management of agricultural land soil restoration, so as to ensure that the environmental risks in the passivation restoration process are controlled.

Response:

In the conclusion of the article, we supplemented the main management and control measures and suggestions to deal with the risks in the whole process, including the supervision suggestions before, during and after the construction, which are consistent with the environmental risks identified in the article.

Reviewer 2 Report

Dear Editor:

Thank you for giving me the opportunity to revise the MS entitled “Whole-Process Risk Management of Soil Amendments for Remediation of Cd- A Review” by Hanlin Feng and his/her colleagues that was submitted to “IJERPH”. The MS submitted is suitable for IJERPH, and some interesting results were showed. However, there are several requirements that have to consider by the authors. In this regard, the following comments are requested to be addressed by the authors:

The English of the paper is readable; however, I would suggest the authors to have it checked preferably by a native English-speaking person to avoid any mistakes. 

Comment 1: Please modify the title. Firstly, Cd is mentioned in the title, but it seems that the center of the full text is not Cd. Secondly, agricultural land is used in the keywords, but it seems that the title and the center of full text is not agricultural land.

Comment 2: You need to add an extra information in the abstract. Kindly focus on presenting the key progress and novelties from the review itself.

Comment 3: The novelty of this literature research should be inserted in the text clearly.

Comment 4: The full text is just a list of some methods, lacking in-depth analysis and discussion.

Comment 5: The full text is a little short, and the number of references is not enough.

Comment 6: The title of the MS is " Whole-Process Risk Management of Soil Amendments for Remediation of Cd- A Review ". However, the author only reviews risk identification in this MS, and has not yet covered how to control risks. Risk management should include four processes at least: risk identification, risk analysis, risk control and risk assessment.

Comments 7: Please carefully check the format of references, such as superscripts, capitalization, journal abbreviations, etc.

For example, Line 275 Line277 Line286 Line 288 Line290 Line294 Line298 Line326 Line 371 Line 376 Line 412

Comment 8: Some references are too old. Authors should try to quote references from the last three or five years.

I would suggest that the authors review and include the following recent studies about soil amendments to improve the manuscript.

1.    Su, R.; Ou, Q.; Wang, H.; Luo, Y.; Dai, X.; Wang, Y.; Chen, Y.; Shi, L., Comparison of phytoremediation potential of Nerium indicum with inorganic modifier calcium carbonate and organic modifier mushroom residue to lead-zinc tailings. Int. J. Environ. Res. Public Health 2022, 19, (16), 10353.

2.    He, L.; Su, R.; Chen, Y.; Zeng, P.; Du, L.; Cai, B.; Zhang, A.; Zhu, H., Integration of manganese accumulation, subcellular distribution, chemical forms, and physiological responses to understand manganese tolerance in Macleaya cordata. Environ Sci Pollut R 2022, 29, (26), 39017-39026. 

Best regards,

Author Response

Response to reviewer 2.

Dear Editor:

Thank you for giving me the opportunity to revise the MS entitled “Whole-Process Risk Management of Soil Amendments for Remediation of Cd- A Review” by Hanlin Feng and his/her colleagues that was submitted to “IJERPH”. The MS submitted is suitable for IJERPH, and some interesting results were showed. However, there are several requirements that have to consider by the authors. In this regard, the following comments are requested to be addressed by the authors:

The English of the paper is readable; however, I would suggest the authors to have it checked preferably by a native English-speaking person to avoid any mistakes. 

Comment 1: Please modify the title. Firstly, Cd is mentioned in the title, but it seems that the center of the full text is not Cd. Secondly, agricultural land is used in the keywords, but it seems that the title and the center of full text is not agricultural land.

Response:

We changed the title, and used heavy metals instead. We also used stabilization instead of agricultural land.

Comment 2: You need to add an extra information in the abstract. Kindly focus on presenting the key progress and novelties from the review itself.

Response:

In the abstract, we supplemented the innovation points and some interesting points of the paper. This article is the first time to identify the environmental risks in the whole process of the application of soil heavy metal stabilization by use amendment, and found that many risks come from technology selection and non-standard use, which can provide support for the safe and effective application of soil improvement technology in the next step.

Comment 3: The novelty of this literature research should be inserted in the text clearly.

Response:

We supplement the innovation of this study in two places. One is in the abstract (which is consistent with comment 2); Second, in the preface, it is emphasized that the comprehensive environmental risk identification of in-situ stabilization technology of heavy metals in farmland soil by using amendments has not been carried out at present. Meanwhile, this study divides environmental risks into three stages, namely, preparation stage, construction process and completion of the project, and according to the characteristics of environmental risks, it can be divided into three types: persistent risk, toxic risk and complex risk.

Comment 4: The full text is just a list of some methods, lacking in-depth analysis and discussion.

Response:

This is a review article, aiming at a comprehensive review of current research. We mainly sorted out the current stage of soil amendments to repair heavy metal contaminated farmland technology problems for a comprehensive identification. As there are few studies on the risk analysis of this technology at the current stage, we have not carried out a more in-depth analysis. Some ideas and extrapolations are given in the conclusion section(lines 85-92).

Secondly , conceptual models are added in appropriate parts of the paper to help readers understand the risk sources and risk processes in the use of soil heavy metal remediation amendments discussed in the paper(line 152).

Comment 5: The full text is a little short, and the number of references is not enough.

Response:

We have added a number of new research literatures, including the status quo of heavy metal pollution in farmland soil in different countries and the research status of risk control in the whole process of soil remediation technology(line 68-74). At the same time, in the section of risk identification, we added some new research literature.

Comment 6: The title of the MS is " Whole-Process Risk Management of Soil Amendments for Remediation of Cd- A Review ". However, the author only reviews risk identification in this MS, and has not yet covered how to control risks. Risk management should include four processes at least: risk identification, risk analysis, risk control and risk assessment.

Response:

In the first part of the paper, we add the research status of risk analysis, evaluation and control. At present, the risk identification and control of soil amendments for remediation of heavy metal pollution in farmland are less, so it is not fully discussed.

Comments 7: Please carefully check the format of references, such as superscripts, capitalization, journal abbreviations, etc.

For example, Line 275 Line277 Line286 Line 288 Line290 Line294 Line298 Line326 Line 371 Line 376 Line 412

Response:

Accepted. The revised literature corresponds to Line 305, Line 309, Line 317, Line 326,  Line 328, Line 326, Line 336, Line 365, Line 424,  Line 458. We also checked other references.

Comment 8: Some references are too old. Authors should try to quote references from the last three or five years.

Response:

We checked the references, and when it’s necessary, we updated some literature published too early.  We also added one of the studies that you suggested(line).

Reviewer 3 Report

This manuscript gives an attempt to review a part of the soil remediation that uses substances to stabilize contaminants. Its English is bad, which makes a bad impression. Not much work on soil remediation is published in scientific literature. The manuscript is written in a balanced manner: it makes problems were hardly problems exist, and make large simplifications. On the other hand it does refer to relevant work.

However the manuscript is not interesting to read. After reading it the reader is not much wiser: most parts of the text are platitudes. Can it be improved?  The work is not clearly focused. Is it about all types of amendments to agricultural soil that have been polluted with heavy metals? The title states that it is about cadmium, then please focus only to this, and indeed mention the practices that help decrease cadmium exposure but unintentionally increase exposure to other pollutants if they are there. In case of organic contaminants it seems more logical add substances or practices to enhance degradation. And in case of non-agricultural soils, many other practices can be used such as covering the soil etc.   

Detailed remarks

37 The first target is to decrease exposure: for soil biota, via drinking water after for leaching and via crop and animal products after uptake by plants, and by inhalation of air as soil dust. Stabilization does not necessarily decrease dust from polluted soils.

Please explain first stabilization or give some examples.

44 “....main application technique...”. Who says so? Please give a reference for such a bold statement. I do not know any member state in the EU who uses these techniques. In n line 116 you write the opposite: “the application of this technology in soil remediation is limited.

60-61 It is still vague what you mean by “this technique”.

1-73 The use of English is far from adequate.

87 -90 These sentences are completely uncomprihensable

90 mcda?

Table 1 MATRIX MCDA, MAHP, AHP-TOPSIS... all these name are not explained.

Area 1. It is the Port of Marghera, a part of the city of  Venice: so it is “Porto Margheta, Venice” not “Porto”,

area 5. It is Iowa not Lowa.

Table 2 UKEPA? It is the Environmental Agency, not a UK version of the US EPA.

HKEPD, USEAP, etc what do these names stand for? The header of the column says: country or region.

145 “often contains PAH”. How can you write this without references, while earlier you mention the “The Biochar Initiative” which has set limit values for PAH. Also the recent EU Fertiliser Product Regulation has set limit values for PAH in biochar.

149 arsenic is not considered a heavy metal.

164 “biochar may cause acidification”. Remarkable, as most biochars have very high pH values. Reading the paper: the pH was increased. So what you write is a wrong citation.

174 “enter the atmospheric environment ...” The text is full these with simple aspects that are hardly very relevant because it is easy to handle. Even a farmer is able to apply lime: you use simple tools to prevent dust. These arguments are very not relevant.

Table 3 gives a nice overview of some literature, but some aspects are so simple: “the material itself contains contaminants”. Of course this is an environmental risk. Is it necessary to make a point of this?

Also Tbale 3 gives a nice overview of the problems and uncertainties. That would be logical if others assume that the effect of the techniques are certain. But I guess that is not the case. So the starting point might also be the various techniques and an evaluation of it effects.

Author Response

(The authors gave the same response as above.)

Round 2

Reviewer 2 Report

In this version, the work is more understandable and presented in a sufficient way. In my opinion, it is suitable for publication.

Author Response

thanks for the reviewer's comments. MDPI have edited the paper carefully for us, and we uploaded the supporting document.

Reviewer 3 Report

After a rejection, normally there is not a second chance. The English in the first version was bad, and it has not been improved. Why not? The manuscript has only been improved based on the comments from the reviewers, but this still does not make it a well-balanced manuscript. The comments were examples, and by repairing only these aspects the manuscript is still not ready.  

After reading the first paragraph the reader still does not know what the problem is and what the remediation is. However, I guess the authors are correct in their assumptions and I have sympathy for their struggle. But it is written in an incomprehensible style. On the one hand they say that there is a technique to decrease the risks of heavy metals in farmland soil, and on the other hand they say that nobody uses these techniques due to various reasons. Their solution is to write a paper about “whole-process risk management”. I think this is not helpful. Why create this contrast?   Why not simply write about remediation techniques for farmland soils and the pros and cons of these techniques?

After reading the first paragraph I concluded that the manuscript is still far from being finished. Detailed remarks about language in the first paragraph to give an impression:

30 “livestock and poultry” Why use this differentiation?

32 “heavy metal in soil can be moved”. This suggests that you can move heavy metals like furniture. You mean that heavy metals can be mobile.

33 “accumulated to” accumulated in

33 “transmission of crops roots” transmission of heavy metals in roots

46 “industry site”. industrial site.

48  “biological activity of heavy metals“. Heavy metals are not biologically active. You mean: remediation should reduce the effects of heavy metals on biological activity.

52 “in particular”??

58 “stabilizaitng”. Please check spelling.

50 “....main application technique...”. I asked: who says so? Now you now give [12,13] as references, but in references [12] I cannot find anything that seems like you statement.

60-61 It is still vague what you mean by “this technique”.

62 “under controllable risks” Strange sentence

64 “Especially for some ....relatively large”. This sentence is incomprehensible

68 “..the Cd content of farmland soil in China is 2.58 times that of EU and 10.75 times that of Australia [20-21].” How do you come up with this? Reference [20] states: “Compared with the reported global average contents, the resulted national mean values for majority elements in China were lower than the global contents, similarly as European areas (Rawlins et al., 2012; Reimann et al., 2012). The average content of Cd in the United States is 0.34 mg/kg (Holmgren et al., 1993), which is higher than the mean value in China.”  

70 “objective conditions” You mean the objective. Do you have any reference that it is there is a policy in the EU of US to restore farmland soil from heavy metals? No, I don’t think this policy exists.  

Author Response

thanks for the reviewer's comments. MDPI have edited the paper carefully for us, and we uploaded the supporting document.

  •  
